# Young dictators—Speaking about oneself decreases generosity in children from two cultural contexts

**Sandra Weltzien**[1¤a]*, **Lauren Marsh**[1¤b], **Patricia Kanngiesser**[2], **Bruce Hood**[1]

**1** Bristol Cognitive Development Centre, School of Experimental Psychology, University of Bristol, Bristol, United Kingdom, **2** School of Psychology, University of Plymouth, Plymouth, United Kingdom

¤a Current address: Faculty of Health Sciences and Social Care, Molde University College, Molde, Norway
¤b Current address: School of Psychology, University of Nottingham, Nottingham, United Kingdom
* sandra.m.weltzien@himolde.no

## Abstract

Sharing of resources is a common feature of human societies. Yet, there is substantial societal variation in children's generosity, and this variation emerges during middle childhood. Societal differences in self-construal orientation may be one factor influencing the ontogeny of generosity. Here, we examine anonymous Dictator Game sharing in 7-and-8-year-olds from two distinct societies: India and the UK (*N* = 180). We used self-construal manipulations to investigate whether priming self- or other-focused conversations would differentially influence children's generosity. There were no differences in generosity between populations. While a significant reduction in generosity was found following self-priming in both societies, other-priming was ineffectual. The findings are discussed in relation to experimental features and the role of anonymity and reputational concerns.

**Data Availability Statement:** All relevant data are within the manuscript and its Supporting Information files.

**Funding:** This research was supported by an Economic and Social Research Council (ESRC)

## Introduction

Most people share resources with others, but to varying degrees across societies and ages (e.g., [1–3]). Developmentally, there appears to be a consistent pattern where young children initially show a bias towards not sharing with others, but are increasingly more likely to share and to share more resources as they get older [4–7]. For example, when asked to share resources with other non-related individuals, pre-schoolers typically self-maximize [2, 3, 8, 9]. However, from middle childhood (7–8 years), children become sensitive to normative information about costly sharing and increasingly conform to their society's generosity norms [3, 4, 5, 9].

Whether we share or not has been linked to the way that we think about our self. Indeed, being "selfish" is a common definition of not sharing. Self-construal can be defined as the collection of thoughts, feelings and actions concerning the self as separate from others, and the self in connection with others [10]. Consequently, one typically separates between independent and interdependent construals of self [11]. It should, however, be noted that independent and interdependent self-construals coexist in all individuals, but can be emphasised and

grant, received by BH (grant no. ES/K010131/1). URL: https://www.ukri.org/councils/esrc/. PK was supported by a Freigeist Fellowship from Volkswagen Foundation (grant no. 89611). URL: https://www.volkswagenstiftung.de/en. The funders had no role in study design, data collection and analysis, decision to publish, or preparation of the manuscript.

**Competing interests:** The authors have declared that no competing interests exist.

accessed to varying degrees [10]. Research on the link between self-construal and generosity in children dates back at least 40 years. For example, when 7- to 12-year-olds were asked to recall happy, sad or neutral events experienced by themselves or with another child, those who recalled sad events involving another, subsequently shared more than those who recalled sad personal events [12]. More recently, studies of pre-schoolers' understanding of the emotional consequences of sharing for themselves versus another recipient child, show that they readily understand the positive emotional benefits of sharing for themselves [13]. Moreover, when pre-schoolers are induced to share in one situation, they become increasingly generous in subsequent sharing situations [14]. This work indicates that that from preschool years and onwards, children are both aware of the personal emotional benefits of sharing and can be induced to be more generous. However, one caveat is that these experimental manipulations often involve public displays of generosity. For example, in the three studies mentioned above [12–14], sharing was carried out in view of an experimenter. Both adults and children behave more generously when others witness their actions (e.g., [15–22]) which suggests that reputational effects should be taken into consideration when considering the development of generosity.

Societal variation in children's prosociality is often linked to differing socialization goals and practices (e.g., [23–26]). For example, Western middle-class families emphasise autonomy and individuality in their child rearing, arguably leading children to develop more independent self-construals [26–33]. Conversely, other societies promote connectedness and group affiliation, arguably fostering more interdependent self-construals [26–28, 33–36]. Thus, while any individual's self-construal includes both self- and other-focused elements, their relative accessibility and salience varies between societies [33, 37] and may play a role in shaping prosocial behaviours.

A good way to test the impact of self-construal orientation on generosity is to experimentally manipulate it. Recent studies have begun to investigate subtle ways of priming self- and other-focus to influence generosity in sharing [38, 39]. In one study by Weltzien et al. [39], British and Indian children took part in semi-structured interviews where they were asked to talk about themselves (self-focus), friends and family (other-focus), or animals (control condition). Following this simple manipulation, the children completed a forced-choice sharing game. Across 12 trials, the children decided between two mutually exclusive options for distributing tokens to the self and to "another child". The tokens would later be exchanged for stickers. It was found that children from both the UK and India became less generous following self-focus. However, following other-focus, only Indian children from traditional extended families increased generosity. No prosocial effect from other-focus was observed in the British sample [39]. This is an interesting finding, which could suggest that sociocultural differences in the prominence of independent and interdependent self-construals may influence the ease with which children are swayed towards generosity or selfishness. In other words, stronger interdependent self-construals in the Indian participants may have rendered them more susceptible to the prime, resulting in a genuine drive towards generosity.

However, research on the effects of self-construal manipulations on generosity in different sociocultural settings is still in its infancy. In the present experiment, we therefore sought to further explore the effects of self and other focus on children's sharing using the same priming manipulation as Weltzien et al. [39], but a different sharing paradigm which offers a wider range of sharing responses than the forced-choice sharing paradigm and can be easily implemented as an anonymous sharing task; namely the Dictator Game. In the Dictator Game, participants are endowed with a number of items and informed that they can keep all, or share some with a recipient [40]. Any sharing thus entails a cost to the participant. The simplicity of the Dictator Game paradigm has made it a popular tool for examining sharing-decisions

across ages and societies and is usually, but not always, implemented as an anonymous task (e.g., [3, 8, 9, 41–43]). Moreover, as the Dictator Game is one-shot, there is no chance of reciprocity from the recipient, and, unlike the forced-choice sharing paradigm which is conducted in full view, anonymity can easily be controlled [8]. The Dictator Game is therefore by many considered ideal for systematically assessing sharing behaviour as it allows experimenters to minimise social influences.

The current study explored sharing behaviour in British and Indian seven- and eight-year-olds by adopting the priming paradigm developed by Weltzien et al [39] for a new task and sample of children. Across societies, middle childhood (seven-eight years) has been identified as an important phase in prosocial development where self-interested behaviour is diminished, and children begin to adhere to the prosocial norms of their society [3, 4, 9]. Similarly, middle childhood represents an important phase in the development of reputational concerns. Young children do not appear to adjust their prosocial behaviours in ways that improve their reputation. However, by middle childhood children begin to reason explicitly about their reputation and, at this age, a concern for own reputation has been shown to guide aspects of children's prosocial acts, such as sharing and fairness [44].

The United Kingdom and India are two societies that have traditionally been characterised as more individualistic and collectivist respectively [45–47], with corresponding higher rates of generosity and fairness found in more collectivist societies, compared to more individualistic societies (e.g., [23, 25, 38, 39, 48, 49]). Based on previous findings [38, 39], we predicted that self-focus would decrease donations in both British and Indian children. Further, if the societal differences in self-construal orientation between our two populations render Indian participants more receptive to the other-focus prime, we would expect to see an increase in donations from the Indian children, but not the British children, following other-focus compared to the control (i.e. we would see an interaction between prime type and society). Finally, in a separate, exploratory analysis we investigated whether family structure (extended vs. nuclear) would influence Indian children's sharing decisions in the three conditions (see Weltzien et al [39] for similar analyses). Traditional Indian families typically harbour three or more generations, including members of the extended family. In such families, "collective responsibility" is often highly valued, with the needs of the family superseding the needs of the individual. Children from extended families, as opposed to Western-style nuclear families, may thus have more salient interdependent self-construals and may therefore be more susceptible to the interdependence manipulation (i.e. we tested for an interaction between prime type and family structure in the Indian sample only).

## Method

### Participants

The Indian sample included 90 seven- and eight-year-olds enrolled in a middle- to upper-class private school in Pune ($M_{age}$ = 94.53 months, $SD_{age}$ = 6.11, range = 84–106 months; 47 females). The British sample included 90 seven- and eight-year-olds enrolled in middle- to upper-class schools in Bristol ($M_{age}$ = 94.16 months, $SD_{age}$ = 7.24, range = 84–107 months; 48 females). Pune is a large-sized urban city in the West of India, with an estimated population of 4 million residents. Main industries include IT and manufacturing. Bristol is a medium-sized urban city in the South-West of England, with an estimated population of 460,000 people. Main employing industries include Retail and Health and Social Care. The city is diverse with 22% non-white British residents. The sample size was specified a priori in keeping with previous research using a similar method and design [39]. An additional six Indian children and one British child took part but were subsequently excluded from analysis because they failed to

pass one or more of the control questions. Indian children completed all of their lessons in English and had a high level of English language proficiency. Testing was therefore conducted in English by the 1st and 2nd authors in both societies. Participants were tested individually in quiet rooms in their respective schools. Informed consent was provided in writing by all parents, and verbal agreement was obtained from all children who participated in the study.

## Procedure

Prior to data collection, the protocol and procedures of this study were approved by the University of Bristol Faculty of Science Human Research Ethics Committee (ethics approval code: 28041636381). Data was collected in Pune in August and September 2016, and in Bristol in March to July 2017.

## Sticker allocation

Each child was given a selection of ten animal stickers and informed that s/he could select six of them to keep. Once the selection was made, the remaining stickers were removed from sight. The child's selected stickers remained in view, but out of reach on the table.

## Priming-procedure

Adopting the priming-procedure of Weltzien et al [39], each child took part in one of three conditions (between-subjects design): self-focus, other-focus or a control condition. The priming tasks were designed as semi-structured interviews. Questions asked during self- and other-focus interviews were constructed from the list of independent and interdependent self-construal primes by Kühnen and Hannover [50].

In the self-priming condition, the experimenter asked a series of questions about the child aimed to prime notions of self (e.g., "what makes you special?" or "how are you different from other people?"). In the other-priming condition, the experimenter asked questions about the child's relationships with significant others to prime notions of relatedness (e.g., "is there anyone in your life that you feel close to?" or "why is it important to have good friends?"). In the control condition (baseline sharing), the experimenter asked questions about animals, and personal pronouns were carefully avoided. For full priming scripts and experimental instructions, see supporting information (S1 File).

## Dictator game

Two envelopes were introduced to the child. One envelope belonged to the child and it was explained that everything put into this envelope could be taken home. The other envelope belonged to "another, unknown child from a different school". The participating child was then told that he or she could give away some of the stickers to this other child, but that he or she did not *have to* give any stickers away. It was further explained that if he or she wished to give away any stickers, then these could be placed into the "other child's" envelope before posting it into a box. The experimenter turned away during the sharing task so that sharing decisions remained fully anonymous. Control questions were included to ensure that children understood the task. Data from seven children (6 Indian, 1 British) who failed either of the control questions were excluded from analysis.

## Data coding and analysis

All statistical analyses were performed using SPSS statistical software version 23.0. The number of stickers shared was converted to a percentage for each child. Child age in months at the date

of testing was calculated and included as a variable in all analyses. This was done to control for any potential variance associated with age within the sample. Preliminary analyses indicated no effects of gender in any of our analyses so this factor was not considered further. Two dependent measures were calculated from the sharing data. 1) The number of children deciding to share in each priming-condition (i.e. sharing nothing vs. sharing one or more stickers) was analysed using a logistic regression with society, condition and their two-way interaction as well as age as predictors. 2) The percentage of stickers shared was analysed with an ANCOVA, followed by Bonferroni-adjusted comparisons. Prime type and society and their two-way interaction were entered as between subject factors, and age in months was entered as a covariate. All summary statistics are presented in Table 1, and the raw data file is available as supporting information (S2 File). Finally, following from Weltzien et al [39], a separate analysis was conducted to explore the effect of family structure on sharing behaviour in the Indian population. The analytical procedure was identical to the procedure described above. However, instead of including culture as a predictor (logistic regression) and between subjects' factor (ANCOVA), the current analysis included family structure (extended vs. nuclear).

Average time spent on the priming interviews did not differ significantly across societies, $F(1, 178) = 1.78$, $p = .18$, $\eta_p^2 = .006$. Moreover, interview times did not differ significantly across conditions in the Indian population, $F(2, 87) = .91$, $p = .41$, $\eta_p^2 = .02$, nor in the British population, $F(2, 87) = .37$, $p = .69$, $\eta_p^2 = .008$. Therefore, this factor will not be considered further. Summary statistics of all priming time data is available as supportive information (S3 File).

## Results

### Decision to share

A test of the full model against a model containing only the intercept was statistically significant, indicating that the predictors together reliably distinguished between sharing and not sharing, $X^2(6) = 23.61$, $p < .001$. The results revealed a trending reduction in willingness to share following self-priming compared to the control ($\beta = -3.54$, $p = .058$). There was no significant difference in decision to share between other-priming and the control ($\beta = -.39$, $p = .854$). Finally, society was not a significant predictor of sharing decision ($\beta = -.66$, $p = .133$), and there was no significant interaction between society and priming condition ($p = .448$). Nevertheless, an inspection of the frequency distribution of donations in each population revealed that the number of zero-sharers (children choosing not to share) was more than twice as high in the Indian population compared to the British population following self-priming (see Fig 1). Moreover, only in the British population did we see examples of sharing exceeding the fairness-principle (i.e., sharing more than half of the endowment).

### Percentage of stickers

On average, Indian children shared a smaller percentage of their stickers ($M_{India} = 25.74\%$, 95% $CI_{India} = [21.99, 29.48]$) compared to British children ($M_{UK} = 29.81\%$, 95% $CI_{UK} = [26.25,$

**Table 1. Summary statistics for sharing data.**

| Prime Type | Society | N | Decision to share | % Shared [95%CI] |
|---|---|---|---|---|
| **Self** | India | 30 | 16 | 14.99 [9.03, 20.97] |
| | UK | 30 | 24 | 23.33 [17.31, 29.36] |
| **Other** | India | 30 | 27 | 30.56 [24.42, 36.69] |
| | UK | 30 | 27 | 30.56 [24.00, 37.11] |
| **Neutral** | India | 30 | 28 | 31.67 [25.48, 37.86] |
| | UK | 30 | 28 | 35.55 [29.72, 41.39] |

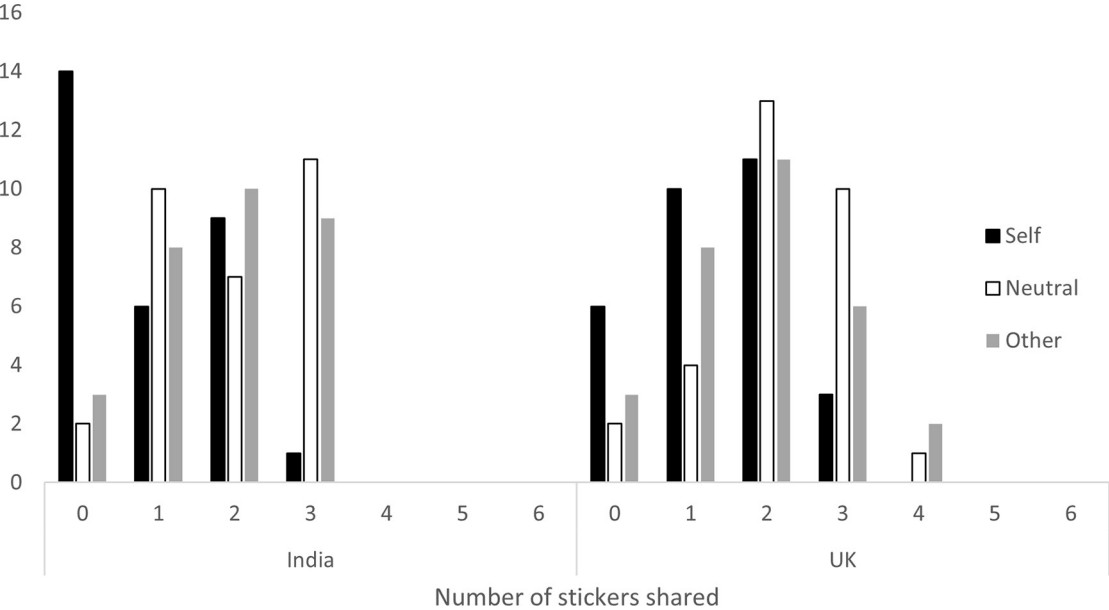

**Fig 1. Frequency distribution of donations across populations and conditions.**

33.38]), but this difference was not significant, $F(1, 173) = 2.77$, $p = .098$, $\eta_p^2 = .02$. No significant interaction was found between priming condition and society, $F(2, 173) = .966$, $p = .383$, $\eta_p^2 = .011$. Priming condition had a significant effect on percentage of stickers shared, $F(2, 173) = 12.86$, $p < .001$, $\eta_p^2 = .13$ (see Fig 2). Across both societies, children shared significantly fewer stickers following self-priming ($M_{self} = 19.17\%$, 95% $CI_{self} = [14.91, 23.42]$), compared to the control ($M_{neutral} = 33.61\%$, 95% $CI_{neutral} = [29.45, 37.77]$), $t(118) = -4.86$, $p < .001$, $d = 0.89$ (Bonferroni corrected), and to other-priming ($M_{other} = 30.56\%$, 95% $CI_{other} = [26.20, 34.91]$), $t(118) = -3.74$, $p < .001$, $d = 0.68$ (Bonferroni corrected). There was no significant difference in

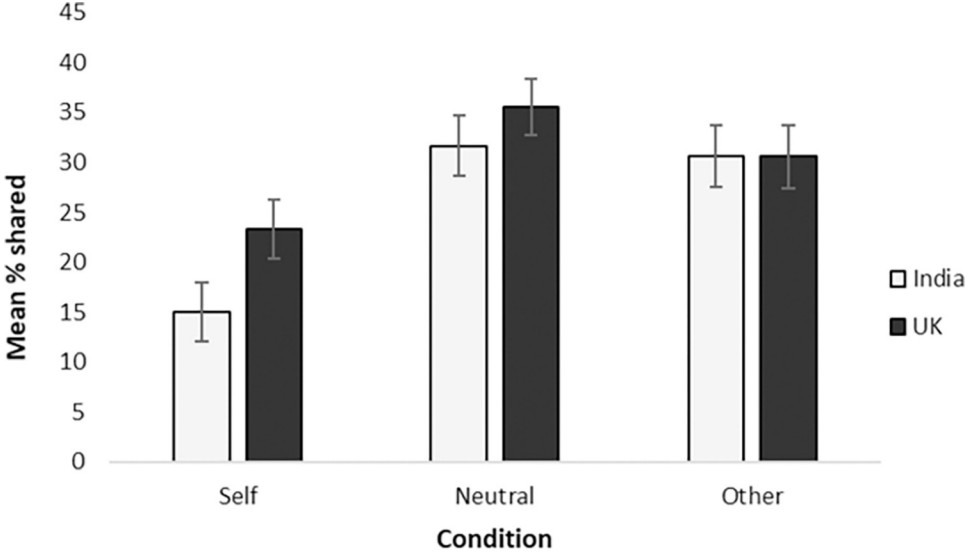

**Fig 2. Mean percentage of stickers shared as a function of prime type.** Error bars show standard errors of the mean.

the percentage of stickers shared following other-priming compared to the control, $t(118) =$ -1.02, $p = .312$, d = 0.19.

## Family structure data, Indian sample

Of the current participants, 48 children lived in traditional extended families, whereas 42 lived in nuclear families. Family structure (extended vs. nuclear) was not a significant predictor of decision to share ($\beta = .85$, $p = .110$). Moreover, there was no effect of family structure on percentage of stickers shared, $F(1, 83) = 1.05$, $p = .309$, $\eta_p^2 = .012$, and no interaction between family structure and condition, $F(2, 83) = .03$, $p = .970$, $\eta_p^2 = .001$.

## Discussion

In the current study, we explored the effects of self- and other-focus on British and Indian children's generosity in a simple Dictator Game. We did not measure existing self-construals in this study, but were interested in how children would react to priming that focused on self or relatedness to others. In both societies, focusing on oneself reduced donations compared to a control condition. This is in keeping with evidence of reductions in sharing, trading and helping following self-focus [38, 51]. Moreover, this finding is consistent with previous cross-societal evidence [39] and extends this research to a different sharing paradigm that allows for more nuanced responses. However, other-focus was ineffectual across both groups. Consequently, the present results diverge from the findings of Weltzien et al [39], who revealed *increased* generosity following identical other-focus in Indian participants from extended families. What could explain these diverging outcomes?

In experimental settings, subtle methodological variations can substantially influence sharing decisions (e.g., [18, 52, 53]). For example, it is well documented that donations in sharing games vary as a function of anonymity. Indeed, both adults and children behave more generously when others witness their actions (e.g., [15–21]) which suggests that reputational effects should be taken into consideration when considering the development of generosity, particularly in middle childhood. Anonymity may be particularly important in cross-societal comparisons of sharing behaviour as reputational concerns and the perceived role and status of the experimenter may differ vastly between societies (e.g., [1, 54]).

The vast majority of previous cross-cultural sharing studies have used tasks taking place in view of the experimenter (e.g., [3, 4, 5, 24, 39, 48, 49, 55]). Thus, while participants often share with an anonymous receiver, the sharing typically takes place in front of a non-anonymous, adult experimenter, and herein lies a potential problem. Whereas the receiver is oblivious to the participants' decision, the experimenter is not. Sharing decisions may therefore reflect culture-specific concerns and expectations regarding the experimenter, rather than simply being a function of social preferences [1]. While participants in the study by Weltzien et al [39] shared with an unidentified other child, the sharing allocations took place in full view of the experimenter. Participants may not be indifferent to the experimenter's impression of their decisions. For example, participants may be concerned with appearing greedy, or wish to promote a reputation of being generous. It seems plausible that these concerns are stronger in societies that promote and emphasize the value of interdependence. The presence of the experimenter may therefore have influenced both sharing decisions overall, and the effects of the priming in the Indian population. For example, other-focus may have boosted reputational concerns in the Indian participants by triggering an awareness of the experimenter's presence. Note that due to the use of different sharing paradigms, it cannot be strictly determined whether differences in findings between Weltzien et al [39] and the current study are due to reputational concerns or methodological differences. Future studies could investigate this

issue by replicating the current paradigm and comparing sharing behaviour in the experimenter's presence and absence, respectively.

A further possible reason for not detecting an influence of the other-focus primes may be that there is a mismatch between the people being discussed in the other-focused interview and the potential benefactor in the Dictator Game. During self-priming, the person being discussed is the child themselves and later in the dictator game, they are themselves the beneficiary of selfish decisions. The other-focused interview, however, centres on people familiar to the child (e.g., friends and family), yet the potential benefactor is someone different—an unknown child. Future studies should therefore explore whether findings will differ if the benefactor of the Dictator Game is one of the people that the participants were prompted to think about during other-priming. The instructions in our current and previous cross-cultural studies [39] to share with an anonymous child from *another* school, may also undermine the willingness of children to share with someone who is potentially regarded as an outgroup member [2]. This would not explain the differences we found between the studies, but it does offer another potential line of cross-cultural investigation to examine how in-group/out-group categories affect children's willingness to share. Finally, it is also possible that self-priming task could lead the children to view the self in a positive way. Indeed, several of the questions in the self-focus interview were of a positive nature, such as "what are you good at?" and "what makes you special?". Thus, in addition to self- focus, the self-priming interview may have primed a distinctly positive view of the self that could have made participants feel entitled to more resources. Whilst this interpretation does not refute the main finding that self-priming significantly reduces sharing, it could provide an alternative account of how self-priming operates.

In our current study, we find that Indian and British children share 31.67% and 35.55% respectively in the control condition. In adults, the DG is subject to cross-cultural variation with varying levels of generosity typically falling somewhere between 20% and 30% of the original stake (see [40, 56] for reviews). In the available studies of children, however, there is more scope for variation. This may be because adult studies have typically used cash, whereas developmental studies have used a range of age-appropriate resources which may be difficult to match across different groups. For example, stickers are appropriate for younger children, but less so for older children. In one study by Harbaugh, Krause, and Liday [57], seven-, nine-, 10-, 14-, 18-year-olds completed a variety of economic games, including the DG. It was found that the older groups, on average, offered between 10% and 20% of their resources, whereas the seven-year-olds offered less than 10%. This is thus a lower rate of sharing than observed in our current study. However, Gummerum et al., [41] found that children as young as five years of age shared 43% of their stake on average, and Benenson et al. [8] found that four-year-olds, on average, donated between 20% and 30% of their endowment, while six- and nine-year-olds gave slightly more.

The UK and India are two societies that are often characterised as more collectivist and individualistic respectively [45] with corresponding higher rates of generosity and fairness found in more collectivist societies, compared to more individualistic societies (e.g., [23, 24, 48, 49]. However, it is increasingly recognised that the individualism/collectivism dichotomy is overly simplistic and, in some ways, both empirically and theoretically limiting (e.g., [58]), and we did not find this pattern in our sample of children from these two societies. Instead, the current findings fall into the line of recent empirical studies that demand a more refined outlook on societal differences in children's prosociality. For example, exploring the sharing behaviour of 5- to 12-year-olds from five diverse societies, Cowell et al [9] found higher rates of generosity in children from more individualistic societies (The US and Canada), compared to children from China, Turkey and South Africa. Further research revealed an aversion to

receiving more than others (advantageous inequity aversion) from 7–8 years of age in children from the U.S., Canada, and Uganda, but not in children from Mexico, Senegal, rural Peru or rural India [55]. Thus, the rejection of a relative self-advantage was not apparent in a majority of the non-Western populations. We suggest that societal variations in reputational concerns and subtle methodological differences could play a critical role in the divergent outcomes that have emerged from cross-societal studies of children's sharing behaviour. A careful re-examination of previous cross-societal studies from the reputation perspective using consistent, anonymous methodologies may therefore be required to disentangle the relationship between reputation and developing prosociality.

While the impact of self- and other-focus on British and Indian children's generosity was the focus of the current study, there are other socioecological differences between the two societies that could influence the results. One such difference is the relative affluence of the two societies. In both India and the UK, we sampled from schools that were mostly attended by children from middle to upper class families. While this ensured that social class was similar within the respective countries, children from UK families may have nevertheless been more affluent when comparing absolute household income. Objective income [8] but also subjective SES may have a different impact on children's resource distributions and their judgements about distributions [59]. However, in the current study, we found no significant differences in generosity between Indian and British participants.

In conclusion, the current findings provide further evidence that self-focus increases children's selfishness in two distinct societies. Yet, other-focus was ineffectual. While it is remarkable that children across societies can so easily be moved toward greater selfishness, the difficulty in moving them toward greater generosity is equally noteworthy. We speculate that differences in reputational concerns may explain some of the divergent outcomes that have emerged from cross-societal studies of children's sharing behaviour.

## Supporting information

**S1 File. Priming scripts and experimental instructions.**
(DOCX)

**S2 File. Raw data.**
(XLSX)

**S3 File. Priming time data.**
(DOCX)

**S4 File. Inclusivity in global research.**
(DOCX)

## Acknowledgments

We wish to thank Dr Jahnavi Sunderarajan for helpful insight and assistance with school recruitment in Pune, India. We are also grateful to all children, parents and schools who participated in the studies and made this research possible.

## Author Contributions

**Conceptualization:** Sandra Weltzien, Bruce Hood.

**Formal analysis:** Sandra Weltzien, Patricia Kanngiesser.

**Funding acquisition:** Bruce Hood.

**Investigation:** Sandra Weltzien, Lauren Marsh.

**Methodology:** Sandra Weltzien, Lauren Marsh, Patricia Kanngiesser, Bruce Hood.

**Project administration:** Sandra Weltzien, Bruce Hood.

**Supervision:** Bruce Hood.

**Validation:** Bruce Hood.

**Visualization:** Sandra Weltzien.

**Writing – original draft:** Sandra Weltzien.

**Writing – review & editing:** Lauren Marsh, Patricia Kanngiesser, Bruce Hood.

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
