## [Decision Letter · Decision Letter 0]

30 Jun 2023

PONE-D-23-07589Young dictators - Speaking about oneself decreases generosity in children from two cultural contextsPLOS ONE

Dear Dr. Weltzien

Thank you for submitting your manuscript to PLOS ONE. After careful consideration, we feel that it has merit but does not fully meet PLOS ONE’s publication criteria as it currently stands. Therefore, we invite you to submit a revised version of the manuscript that addresses the points raised during the review process. We have received 2 reports on your paper. Both reviewers are positive about the chances of your paper and both recommend minor revision.

We look forward to receiving your revised manuscript.

Kind regards,

Jaume Garcia-Segarra

Academic Editor

PLOS ONE

Journal Requirements:

Reviewers' comments:

Reviewer's Responses to Questions

**Comments to the Author**

1. Is the manuscript technically sound, and do the data support the conclusions?

Reviewer #1: Yes

Reviewer #2: Yes

2. Has the statistical analysis been performed appropriately and rigorously? 

Reviewer #1: Yes

Reviewer #2: Yes

3. Have the authors made all data underlying the findings in their manuscript fully available?

Reviewer #1: Yes

Reviewer #2: Yes

4. Is the manuscript presented in an intelligible fashion and written in standard English?

Reviewer #1: Yes

Reviewer #2: Yes

5. Review Comments to the Author

Reviewer #1: This manuscript presents the results of a study on how self-construal priming affects children’s generosity in India and the UK. Self-priming was found to reduce sharing relative to no priming, while Other priming and no priming produced similar levels of sharing. These findings constituted a partial replication of prior work; they provide additional support for the claim that self-priming reduces sharing, while the effects of other priming appear to be more context-dependent and did not emerge in the current sample. In terms of the cross-cultural findings, results were mostly similar in India and the UK, although differences may have emerged with a slightly larger sample size.

Overall, I enjoyed this paper. I found it to be well-written and succinct, with useful summaries of prior work on this topic. I enjoyed the authors’ mention of the complexities of “individualistic vs. collectivist” distinction, which is too often presented in the literature as straightforward and unproblematic. The methods were clear and the results rather straightforward, nicely replicating past work on how self-priming affects children’s generosity. The overall contribution is not extremely novel in light of prior research (please note that I am aware that novelty per se is not emphasized for this particular review process), but it still helps advance the field in the terms of testing the boundary conditions of prior effects. While I do have several suggestions for improvement, my overall impression of this contribution is positive. My proposed changes are fairly straightforward and should be addressable in a revision. I have a general concern that the study is under-powered (and the lack of an a prior power analysis is a shortcoming) for examining certain effects (e.g., sample sizes for the Indian sample for the different family structures were quite small). That said, these concerns can easily be addressed via post-hoc sensitivity analyses and more discussion of how confidently we can interpret null findings as such. Additional data collection is not needed. I shall outline my suggestions in the next section of this review.

Page 2: “where young children initially show a bias towards not sharing with others, but are increasingly more likely to do so” – The authors’ focus is on whether children share or not, which makes the choice to share seem binary rather than continuous. The discussion of this topic would benefit from briefly noting that children also share more resources when they choose to share as they get older (i.e., the total amount shared increases with age), and not just that they are more likely to share as they get older.

Page 2, last paragraph: The second part of this paragraph focuses on preschoolers, but the age range of the present study does not include preschoolers (I view “preschoolers” as 5 years old or younger). The authors could briefly add a statement that presumably these effects are true of older children as well, to make these findings more directly relevant to the age range of the current study.

“However, one caveat is that these experimental manipulations often involve public displays of generosity.” – Particularly because the private/public distinction wound up being important in the current study, in this section and others, I would appreciate knowing which studies involved public versus private displays of generosity. Of references 10- 12, were all public? And for the Dictator Game studies that follow (when discussed later on in the paper), which involved public versus private donations?

It is worth noting that the study more or less assumes that children in the UK and India have different self-construal orientations, which seems plausible, but the orientations themselves are never directly tested. Could they be tested/are there methods to do so? This could be a topic for the Discussion.

Page 3: “subsequent forced-choice sharing paradigm” – can the authors share a bit more about this? Was it a “mini Dictator Game?” Just a few sentences of detail would be helpful so the readers can get a clearer sense of how methods diverged across studies (it is clear the the current study’s traditional DG is different, but different compared to what?).

Page 5:

A very brief (sentence or two) justification of the age range would be useful. (For instance, was it tied to the original Weltzien at al age range? What about reputation-based concerns for this age range?) I agree based on my knowledge of this literature that 7- and 8-year-olds are suitable, but it’s not entirely clear from the manuscript itself why they were chosen.

Sample size: It is a bit unfortunate that the sample size was set based solely on convention rather than after a power analysis. One solution would be to present results of sensitivity analyses or post-hoc power analyses in this section, e.g., given the relevant sample sizes, what effect sizes would be detected with 80% power? In other words, for the UK compared to India analyses, for instance: how large would the effect have to be to detect between-site differences, etc., given the sample sizes tested? The relevant analyses here are 1) self focus compared to control, in each location separately 2) a condition by site interaction (other focus is expected to boost sharing only in India) 3) nuclear vs. extended family, following other-focus, in India alone; my summary is based on the predictions mentioned on page 4. My overall sense is that the study makes an important contribution, and the self vs. control condition effects are compelling, but that the study is likely underpowered to detect effects 2) and 3) and that the results should be qualified in the Discussion accordingly. In other words, only with very strong effects would a significant interaction and a family-based analysis be detectable given the cell sizes (the cell sizes for 3) are particularly small). The null effects could be considered less as definitive nulls and more as open questions. Shortcomings surrounding the lack of prior power analyses, and limits of conclusions regarding null findings in light of sample size, should be address as limitations in the Discussion. For instance, “As noted, there were no significant differences in generosity between Indian and

British participants. However, we note that our sample size would have been insufficient to detect a small-to-medium effect of location … ” (for example, for modifying page 10)

Page 8: On Page 4, it seemed as though the authors predicted significant effects of self vs. control for each site individually (not just combined across site), but site-specific results are not reported. To make the results consistent with the predictions, I’d suggest reporting the results separately for India and the UK in addition to reporting combined analyses; although I recognize the lack of a significant interaction, I think the predictions are sufficient justification to present analyses for each site.

Page 9: The family structure results, as currently reported, do not quite align with the predictions. In addition to the analyses already presented, please show results for the other condition only, comparing extended vs. nuclear family results (“we predicted higher levels of generosity in Indian children from extended families compared to Indian children from nuclear families following other-focus.”)

Page 9 – The manuscript notes, “reputational effects should be taken into consideration when considering the development of generosity.” There is a missing part of the developmental story here; that concern with reputation is minimal in early childhood but should be operative within the age range of the study. Engelmann & Rapp, 2018, provides a review; some discussion of reputational concerns as changing with development would be useful.

Discussion: One limitation of the general experimental approach is that the self- and other-focus interview questions are not very closely equated. In other words, it’s not the case that the same questions are asked but regarding different targets (for example, “can you think of a time when something good happened to you?” compared to “can you think of a time when something good happened to a friend?”); instead, the content of the questions themselves is quite different, and not just the targets. For instance, “what makes you special?” which is posed in the self-focus interview, has no analog in the other-focus interview. It seems as though the self-focus condition might not just prime a self focus but cause one to view the self in a positive way specifically, while the same is not true of the other-focus condition regarding others. To be clear, I do not think this issue is mostly responsible for the self-focus condition differences, but it should be mentioned as a limitation: namely, that addition to priming self-focus, the self-focus condition may have also primed a distinctly positive view of the self that could have made participants feel entitled to more resources.

Discussion: The role of SES/family income warrants some space in the Discussion. I appreciate the characterization of both schools as “middle- to upper-class,” but due to differences in standard of living in the two countries, the UK school was likely more affluent. There is some work suggesting that family income in and of itself could affect children’s Dictator Game sharing (e.g., Benenson et al), although the direction of the effect is disputed. A brief mention that affluence, in addition to the favored discussion of specifically cultural factors, could have affected the results would create a more complete picture of the relevant issues for the readers.

Readers might be curious as to how the rates of sharing in the current study compare to those of other studies with similar methods and ages; are there any that can be summarized briefly in the Discussion?

Page 11- typo: “For example, Moreover,”

Reviewer #2: 1. I’m a bit curious about the link between the Weltzien et al. (2019) paper and the current study. Was this aimed to be more of a follow-up to that study to see if those patterns would extend to the Dictator Game? Is this an entirely new sample collected at a different date? Some of my additional comments / suggestions are contingent upon knowing the answer to this question.

2. Was there an a priori decision to make the allocation task private (e.g. have the experimenter turn around) and a prediction that this would influence sharing? Again, I think the history and provenance of this study versus the Weltzien (2019) study and the a priori motivations would be helpful for providing context & highlighting the unique contributions of this study.

3. Was there any sort of manipulation check, either in this study or a previous study, to ensure that the primes had the intended effects?

4. It would be helpful to have a clear definition of “self-construal orientation” or some explanation of what it is in both the abstract and in the introduction.

5. I would replace references to “universal patterns” with something like “regularities”.

6. Figure captions should include an explanation of what the error bars represent (e.g. standard deviation, standard errors, etc.)

Dorsa Amir

UC Berkeley

6. PLOS authors have the option to publish the peer review history of their article (what does this mean?). If published, this will include your full peer review and any attached files.

Reviewer #1: No

Reviewer #2: **Yes: **Dorsa Amir

---

## [Author Response · Author response to Decision Letter 0]

4 Dec 2023

Reply to reviewers.

We would like to thank the reviewers for their positive feedback and for their helpful comments. Please see our replies to each of the reviewer’s comments below (in italics). 

Reviewer 1: 

This manuscript presents the results of a study on how self-construal priming affects children’s generosity in India and the UK. Self-priming was found to reduce sharing relative to no priming, while Other priming and no priming produced similar levels of sharing. These findings constituted a partial replication of prior work; they provide additional support for the claim that self-priming reduces sharing, while the effects of other priming appear to be more context-dependent and did not emerge in the current sample. In terms of the cross-cultural findings, results were mostly similar in India and the UK, although differences may have emerged with a slightly larger sample size.

Overall, I enjoyed this paper. I found it to be well-written and succinct, with useful summaries of prior work on this topic. I enjoyed the authors’ mention of the complexities of “individualistic vs. collectivist” distinction, which is too often presented in the literature as straightforward and unproblematic. The methods were clear and the results rather straightforward, nicely replicating past work on how self-priming affects children’s generosity. The overall contribution is not extremely novel in light of prior research (please note that I am aware that novelty per se is not emphasized for this particular review process), but it still helps advance the field in the terms of testing the boundary conditions of prior effects. While I do have several suggestions for improvement, my overall impression of this contribution is positive. My proposed changes are fairly straightforward and should be addressable in a revision. I have a general concern that the study is under-powered (and the lack of an a prior power analysis is a shortcoming) for examining certain effects (e.g., sample sizes for the Indian sample for the different family structures were quite small). That said, these concerns can easily be addressed via post-hoc sensitivity analyses and more discussion of how confidently we can interpret null findings as such. Additional data collection is not needed. I shall outline my suggestions in the next section of this review.

Our Response: Thank you for your positive feedback and your suggestions.

Page 2: “where young children initially show a bias towards not sharing with others but are increasingly more likely to do so” – The authors’ focus is on whether children share or not, which makes the choice to share seem binary rather than continuous. The discussion of this topic would benefit from briefly noting that children also share more resources when they choose to share as they get older (i.e., the total amount shared increases with age), and not just that they are more likely to share as they get older.

Our Response: Thank you for this suggestion. We now mention the increase in total amount shared and reference a recent cross-societal study which found this increase across 12 countries. Specifically, we have reworded the sentence as follows:

“Developmentally, there appears to be a consistent pattern where young children initially show a bias towards not sharing with others, but are increasingly more likely to share and to share more resources as they get older [4-7].”

Page 2, last paragraph: The second part of this paragraph focuses on preschoolers, but the age range of the present study does not include preschoolers (I view “preschoolers” as 5 years old or younger). The authors could briefly add a statement that presumably these effects are true of older children as well, to make these findings more directly relevant to the age range of the current study.

 Our response: Thank you for this comment. We have now amended the text to clarify this point. Specifically, we write:

“Moreover, when pre-schoolers are induced to share in one situation, they become increasingly generous in subsequent sharing situations [14]. This work indicates that from preschool years and onwards, children are both aware of the personal emotional benefits of sharing and can be induced to be more generous”.

“However, one caveat is that these experimental manipulations often involve public displays of generosity.” – Particularly because the private/public distinction wound up being important in the current study, in this section and others, I would appreciate knowing which studies involved public versus private displays of generosity. Of references 10- 12, were all public? And for the Dictator Game studies that follow (when discussed later on in the paper), which involved public versus private donations?

 Our response: Yes, sharing in studies 10-12 (now 12-14) was indeed done in public (in view of an experimenter). We now specify this on page 3, where we write: 

“However, one caveat is that these experimental manipulations often involve public displays of generosity. For example, in the three studies mentioned above [12-14], sharing was carried out in view of an experimenter". 

For the dictator games studies cited later in the introduction (p. 4), some were done anonymously (e.g. references 8, 9, 49), for others it was not entirely clear from the methods whether sharing was done in view of the experimenter or not. We have reworded the sentences as follows:

“The simplicity of the Dictator Game paradigm has made it a popular tool for examining sharing-decisions across ages and societies and is usually, but not always, implemented as an anonymous task (e.g., [3, 8, 9, 41, 42, 43]).”

It is worth noting that the study more or less assumes that children in the UK and India have different self-construal orientations, which seems plausible, but the orientations themselves are never directly tested. Could they be tested/are there methods to do so? This could be a topic for the Discussion.

Our response: Thank you for this suggestion. We have added to the discussion that our study did not focus on existing self-construals but on the effect of priming self vs. others. We now write on p. 10: 

“We did not measure existing self-construals in this study, but were interested in how children would react to priming that focused on self or relatedness to others.”

Page 3: “subsequent forced-choice sharing paradigm” – can the authors share a bit more about this? Was it a “mini Dictator Game?” Just a few sentences of detail would be helpful so the readers can get a clearer sense of how methods diverged across studies (it is clear the the current study’s traditional DG is different, but different compared to what?).

 Our response: Thank you for this suggestion. We have now included a brief description of the forced-choice sharing game on p. 3. Specifically, we write: 

“In one study by Weltzien et al. [39], British and Indian children took part in semi-structured interviews where they were asked to talk about themselves (self-focus), friends and family (other-focus), or animals (control condition). Following this simple manipulation, the children completed a forced-choice sharing game. Across 12 trials, the children decided between two mutually exclusive options for distributing tokens to the self and to “another child”. The tokens would later be exchanged for stickers. It was found that children from both the UK and India became less generous following self-focus. However, following other-focus, only Indian children from traditional extended families increased generosity”.

Page 5: A very brief (sentence or two) justification of the age range would be useful. (For instance, was it tied to the original Weltzien at al age range? What about reputation-based concerns for this age range?) I agree based on my knowledge of this literature that 7- and 8-year-olds are suitable, but it’s not entirely clear from the manuscript itself why they were chosen.

Our response: We agree that the article would benefit from a brief justification of the chosen age range. On page 4, we now write: 

“Across societies, middle childhood (seven-eight years) has been identified as an important phase in prosocial development where self-interested behaviour is diminished, and children begin to adhere to the prosocial norms of their society [3, 4, 9]. Similarly, middle childhood represents an important phase in the development of reputational concerns. Young children do not appear to adjust their prosocial behaviours in ways that improve their reputation. However, by middle childhood children begin to reason explicitly about their reputation, and at this age, a concern for own reputation has been shown to guide aspects of children’s prosocial acts, such as sharing and fairness [44]”.

Sample size: It is a bit unfortunate that the sample size was set based solely on convention rather than after a power analysis. One solution would be to present results of sensitivity analyses or post-hoc power analyses in this section, e.g., given the relevant sample sizes, what effect sizes would be detected with 80% power? In other words, for the UK compared to India analyses, for instance: how large would the effect have to be to detect between-site differences, etc., given the sample sizes tested? The relevant analyses here are 1) self focus compared to control, in each location separately 2) a condition by site interaction (other focus is expected to boost sharing only in India) 3) nuclear vs. extended family, following other-focus, in India alone; my summary is based on the predictions mentioned on page 4. My overall sense is that the study makes an important contribution, and the self vs. control condition effects are compelling, but that the study is likely underpowered to detect effects 2) and 3) and that the results should be qualified in the Discussion accordingly. In other words, only with very strong effects would a significant interaction and a family-based analysis be detectable given the cell sizes (the cell sizes for 3) are particularly small). The null effects could be considered less as definitive nulls and more as open questions. Shortcomings surrounding the lack of prior power analyses, and limits of conclusions regarding null findings in light of sample size, should be address as limitations in the Discussion. For instance, “As noted, there were no significant differences in generosity between Indian and British participants. However, we note that our sample size would have been insufficient to detect a small-to-medium effect of location … ” (for example, for modifying page 10)

Our response: We did not carry out a priori power analyses. However, post-hoc power analyses based on observed power do not provide any additional information over the one already contained in the significance levels of our tests, as observed power is a 1:1 function of the p-value (see, Hoenig & Heisey, 2001, for a detailed explanation). Instead, one recommended approach is to report confidence intervals for all tests (e.g., Hoenig & Heisy, 2001; Yuan & Maxwell, 2005) which we do throughout the manuscript.

References:

Hoenig, J. M., & Heisey, D. M. (2001). The abuse of power: the pervasive fallacy of power calculations for data analysis. The American Statistician, 55(1), 19-24. https://doi.org/10.1198/000313001300339897

Yuan, K.-H., & Maxwell, S. (2005). On the post hoc power in testing mean differences. Journal of Educational and Behavioral Statistics, 30(2), 141–167. https://doi.org/10.3102/10769986030002141

Page 8: On Page 4, it seemed as though the authors predicted significant effects of self vs. control for each site individually (not just combined across site), but site-specific results are not reported. To make the results consistent with the predictions, I’d suggest reporting the results separately for India and the UK in addition to reporting combined analyses; although I recognize the lack of a significant interaction, I think the predictions are sufficient justification to present analyses for each site.

Our response: The study was set up to test a 2x3 design and examine interactions between prime type and society. We have now clarified this in the introduction on p.5, where we write:

“Further, if the societal differences in self-construal orientation between our two populations render Indian participants more receptive to the other-focus prime, we would expect to see an increase in donations from the Indian children, but not the British children, following other-focus compared to the control (i.e. we would see an interaction between prime type and society).”

Page 9: The family structure results, as currently reported, do not quite align with the predictions. In addition to the analyses already presented, please show results for the other condition only, comparing extended vs. nuclear family results (“we predicted higher levels of generosity in Indian children from extended families compared to Indian children from nuclear families following other-focus.”)

Our response: Apologies, the wording in the introduction was not entirely clear. We have reworded the passage as follows: 

“Finally, in a separate, exploratory analysis we investigated whether family structure (extended vs. nuclear) would influence Indian children’s sharing decisions in the three conditions (see Weltzien et al [39] for similar analyses). Traditional Indian families typically harbour three or more generations, including members of the extended family. In such families, “collective responsibility” is often highly valued, with the needs of the family superseding the needs of the individual. Children from extended families, as opposed to Western-style nuclear families, may thus have more salient interdependent self-construals and may therefore be more susceptible to the interdependence manipulation (i.e. we tested for an interaction between prime type and family structure in the Indian sample only).”

Page 9 – The manuscript notes, “reputational effects should be taken into consideration when considering the development of generosity.” There is a missing part of the developmental story here; that concern with reputation is minimal in early childhood but should be operative within the age range of the study. Engelmann & Rapp, 2018, provides a review; some discussion of reputational concerns as changing with development would be useful.

Our response: We agree with this point and thank the reviewer for the suggested reference. As described above, we now include a brief description of reputational concerns as changing with development on p. 4 in relation to the justification of the chosen age-range (see our earlier response for details). We have also slightly amended the sentence on p. 10:

“Indeed, both adults and children behave more generously when others witness their actions (e.g., [15-21]) which suggests that reputational effects should be taken into consideration when considering the development of generosity, particularly in middle childhood. “ 

Discussion: 

1) One limitation of the general experimental approach is that the self- and other-focus interview questions are not very closely equated. In other words, it’s not the case that the same questions are asked but regarding different targets (for example, “can you think of a time when something good happened to you?” compared to “can you think of a time when something good happened to a friend?”); instead, the content of the questions themselves is quite different, and not just the targets. For instance, “what makes you special?” which is posed in the self-focus interview, has no analog in the other-focus interview. 

Our response: It is indeed true that the self- and other-focus interview questions are not completely equated. We wished to adopt the priming procedure used in Weltzien et al [39], and therefore used exactly the same interview questions. This is stated on p 6. The priming questions were not matched perfectly because the other-condition aimed to prime interdependence, not simply a focus on someone else. Questions such as “what makes your friend special” were therefore not included, and instead the focus was placed on questions that would induce feelings of belonging and relatedness. 

2) It seems as though the self-focus condition might not just prime a self focus but cause one to view the self in a positive way specifically, while the same is not true of the other-focus condition regarding others. To be clear, I do not think this issue is mostly responsible for the self-focus condition differences, but it should be mentioned as a limitation: namely, that addition to priming self-focus, the self-focus condition may have also primed a distinctly positive view of the self that could have made participants feel entitled to more resources.

Our response: Thank you for making this observation. We have now added the following paragraph to the Discussion on p. 11: 

“Finally, it is also possible that self-priming task could lead the children to view the self in a positive way. Indeed, several of the questions in the self-focus interview were of a positive nature, such as “what are you good at?” and “what makes you special?”. Thus, in addition to self- focus, the self-priming interview may have primed a distinctly positive view of the self that could have made participants feel entitled to more resources. Whilst this interpretation does not refute the main finding that self-priming significantly reduces sharing, it could provide an alternative account of how self-priming operates.”

Discussion: The role of SES/family income warrants some space in the Discussion. I appreciate the characterization of both schools as “middle- to upper-class,” but due to differences in standard of living in the two countries, the UK school was likely more affluent. There is some work suggesting that family income in and of itself could affect children’s Dictator Game sharing (e.g., Benenson et al), although the direction of the effect is disputed. A brief mention that affluence, in addition to the favored discussion of specifically cultural factors, could have affected the results would create a more complete picture of the relevant issues for the readers.

Our response: We agree that the potential influence of differences in affluence between the testing sites warrants space in the discussion, and have now added the following paragraph on p. 13:

“While the impact of self- and other-focus on British and Indian children’s generosity was the focus of the current study, there are other socioecological differences between the two societies that could influence the results. One such difference is the relative affluence of the two societies. In both India and the UK, we sampled from schools that were mostly attended by children from middle to upper class families. While this ensured that social class was similar within the respective countries, children from UK families may have nevertheless been more affluent when comparing absolute household income. Objective income [8] but also subjective SES may have a different impact on children's resource distributions and their judgements about distributions [59]. However, in the current study we found no significant differences in generosity between Indian and British participants.

Readers might be curious as to how the rates of sharing in the current study compare to those of other studies with similar methods and ages; are there any that can be summarized briefly in the Discussion?

Our response: Thank you for this suggestion. We now provide a brief summary on page 12 of the discussion:

“In our current study, we find that Indian and British children share 31.67% and 35.55% respectively in the control condition. In adults, the DG is subject to cross-cultural variation with varying levels of generosity typically falling somewhere between 20% and 30% of the original stake (see [40, 56] for reviews). In the available studies of children, however, there is more scope for variation. This may be because adult studies have typically used cash, whereas developmental studies have used a range of age-appropriate resources which may be difficult to match across different groups. For example, stickers are appropriate for younger children, but less so for older children. In one study by Harbaugh, Krause, and Liday [57], seven-, nine-, 10-, 14-, 18-year-olds completed a variety of economic games, including the DG. It was found that the older groups, on average, offered between 10% and 20% of their resources, whereas the seven-year-olds offered less than 10%. This is thus a lower rate of sharing than observed in our current study. However, Gummerum et al., [41] found that children as young as five years of age shared 43% of their stake on average, and Benenson et al. [8] found that four-year-olds, on average, donated between 20% and 30% of their endowment, while six- and nine-year-olds gave slightly more.

Page 11- typo: “For example, Moreover,”

Our response: Thank you for pointing out this mistake. It has now been corrected. 

Reviewer 2: 

1. I’m a bit curious about the link between the Weltzien et al. (2019) paper and the current study. Was this aimed to be more of a follow-up to that study to see if those patterns would extend to the Dictator Game? Is this an entirely new sample collected at a different date? Some of my additional comments / suggestions are contingent upon knowing the answer to this question.

Our response: The current study was indeed a follow up to the Weltzien et al. (2019) study to explore the effects of the same priming manipulation on sharing in a different sharing game (the Dictator Game). This is explained on page 4, where we now further specify that the current study used the same priming manipulation as Weltzien et al. (2019), but a different sharing paradigm. Specifically, we write: 

“In the present experiment, we therefore sought to further explore the effects of self and other focus on children’s sharing using the same priming manipulation as Weltzien et al. [39], but a different sharing paradigm which offers full anonymity and a wider range of sharing responses than the forced-choice sharing paradigm; namely the Dictator Game”.

The current study did indeed test an entirely new sample. The data for the two studies were collected close in time, but at different dates. We have now clarified this in the introduction on page 4:

“The current study explored sharing behaviour in British and Indian seven- and eight-year-olds by adopting the priming paradigm developed by Weltzien et al [39] for a new task and sample of children.”

2. Was there an a priori decision to make the allocation task private (e.g. have the experimenter turn around) and a prediction that this would influence sharing? Again, I think the history and provenance of this study versus the Weltzien (2019) study and the a priori motivations would be helpful for providing context & highlighting the unique contributions of this study.

Our response: Yes, it was an a priori decision to make the allocation task private. As stated on page 4 (see extract above), we had two main motivations for choosing the Dictator Game: 1) it offers a wider range of sharing responses than the forced-choice sharing paradigm, and 2) it offers full anonymity. 

We also return to this topic in the discussion on p. 10-11, where we write: 

“The vast majority of previous cross-cultural sharing studies have used tasks taking place in view of the experimenter (e.g., [3, 4, 5, 24, 39, 48, 49, 55]). Thus, while participants often share with an anonymous receiver, the sharing typically takes place in front of a non-anonymous, adult experimenter, and herein lies a potential problem. Whereas the receiver is oblivious to the participants’ decision, the experimenter is not. Sharing decisions may therefore reflect culture-specific concerns and expectations regarding the experimenter, rather than simply being a function of social preferences [1]. While participants in the study by Weltzien et al [39] shared with an unidentified other child, the sharing allocations took place in full view of the experimenter. Participants may not be indifferent to the experimenter’s impression of their decisions. For example, participants may be concerned with appearing greedy, or wish to promote a reputation of being generous. It seems plausible that these concerns are stronger in societies that promote and emphasize the value of interdependence. The presence of the experimenter may therefore have influenced both sharing decisions overall, and the effects of the priming in the Indian population. For example, other-focus may have boosted reputational concerns in the Indian participants by triggering an awareness of the experimenter’s presence.”

3. Was there any sort of manipulation check, either in this study or a previous study, to ensure that the primes had the intended effects?

Our response: We would argue that the outcome measures are the measure of whether the primes had an effect. The questions we asked during the self- and other-focus interviews were created based on the list of independent self-construal primes (e.g., unique, alone) and interdependent self-construal primes (e.g., group, cooperate) developed by Kühnen and Hannover [50]. Kuhnen and Hannover [50] successfully used these self-construal primes in scrambled sentence tasks to manipulate perceived self–other similarity in adults, which increased after interdependence priming and decreased after independence priming. This suggests that interdependence-priming works to induce notions of relatedness in adults.

4. It would be helpful to have a clear definition of “self-construal orientation” or some explanation of what it is in both the abstract and in the introduction.

Our response: Thank you for this suggestion. The following definition has now been added to the second paragraph of the introduction on p.2:

“Self-construal can be defined as the collection of thoughts, feelings and actions concerning the self as separate from others, and the self in connection with others [10]. Consequently, one typically separates between independent and interdependent construals of self [11]. It should, however, be noted that independent and interdependent self-construals coexist in all individuals, but can be emphasised and accessed to varying degrees [10].

5. I would replace references to “universal patterns” with something like “regularities”.

Our response: “Universal patterns” has now been replaced by “consistent pattern”. Specifically, we now write on p. 2:

“Developmentally, there appears to be a consistent pattern where young children initially show a bias towards not sharing with others, but are increasingly more likely to share and to share more resources as they get older [4-7].”

6. Figure captions should include an explanation of what the error bars represent (e.g. standard deviation, standard errors, etc.)

Our response: The figure caption already includes a description of what the error bars represent. Specifically, on page 9 we write: 

“Fig 2. Mean percentage of stickers shared as a function of prime type. Error bars show standard errors of the mean”.

---

## [Decision Letter · Decision Letter 1]

23 Feb 2024

Young dictators - Speaking about oneself decreases generosity in children from two cultural contexts

PONE-D-23-07589R1

Dear Dr. Weltzien,

We’re pleased to inform you that your manuscript has been judged scientifically suitable for publication and will be formally accepted for publication once it meets all outstanding technical requirements.

Kind regards,

Jaume Garcia-Segarra

Academic Editor

PLOS ONE

Additional Editor Comments (optional):

All concerns raised in the first submission have been addressed and clarified.

Reviewers' comments:

Reviewer's Responses to Questions

**Comments to the Author**

1. If the authors have adequately addressed your comments raised in a previous round of review and you feel that this manuscript is now acceptable for publication, you may indicate that here to bypass the “Comments to the Author” section, enter your conflict of interest statement in the “Confidential to Editor” section, and submit your "Accept" recommendation.

Reviewer #1: All comments have been addressed

2. Is the manuscript technically sound, and do the data support the conclusions?

Reviewer #1: Yes

3. Has the statistical analysis been performed appropriately and rigorously? 

Reviewer #1: Yes

4. Have the authors made all data underlying the findings in their manuscript fully available?

Reviewer #1: (No Response)

5. Is the manuscript presented in an intelligible fashion and written in standard English?

Reviewer #1: Yes

6. Review Comments to the Author

Reviewer #1: (No Response)

7. PLOS authors have the option to publish the peer review history of their article (what does this mean?). If published, this will include your full peer review and any attached files.

Reviewer #1: No

---

## [Editor Report · Acceptance letter]

28 Feb 2024

PONE-D-23-07589R1 

PLOS ONE

Dear Dr. Weltzien, 

I'm pleased to inform you that your manuscript has been deemed suitable for publication in PLOS ONE. Congratulations! Your manuscript is now being handed over to our production team.

Kind regards, 

on behalf of

Dr. Jaume Garcia-Segarra 

Academic Editor

PLOS ONE